# Identifying fetal yawns based on temporal dynamics of mouth openings: A preterm neonate model using support vector machines (SVMs)

Damiano Menin[1], Angela Costabile[2], Flaviana Tenuta[2], Harriet Oster[3,4], Marco Dondi [1]*

**1** Dipartimento di Studi Umanistici, Università degli Studi di Ferrara, Ferrara, Italy, **2** Dipartimento di Culture, Educazione e Società, Università della Calabria, Cosenza, Italy, **3** School of Professional Studies, New York University, New York City, New York, United States of America, **4** Department of Psychology, Hunter College, City University of New York, New York City, New York, United States of America

* marco.dondi@unife.it

**Data Availability Statement:** Data are available as supporting information.

**Funding:** The authors received no specific funding for this work.

## Abstract

Fetal yawning is of interest because of its clinical, developmental and theoretical implications. However, the methodological challenges of identifying yawns from ultrasonographic scans have not been systematically addressed. We report two studies that examined the temporal dynamics of yawning in preterm neonates comparable in developmental level to fetuses observed in ultrasound studies (about 31 weeks PMA). In Study 1 we tested the reliability and construct validity of the only quantitative measure for identifying fetal yawns in the literature, by comparing its scores with a more detailed behavioral coding system (The System for Coding Perinatal Behavior, SCPB) adapted from the comprehensive, anatomically based Facial Action Coding System for Infants and Young Children (Baby FACS). The previously published measure yielded good reliability but poor specificity, resulting in over-representation of yawns. In Study 2 we developed and tested a new machine learning system based on support vector machines (SVM) for identifying yawns. The system displayed excellent specificity and sensitivity, proving it to be a reliable and valid tool for identifying yawns in fetuses and neonates. This achievement represents a first step towards a fully automated system for identifying yawns in the perinatal period.

## Introduction

Fetal yawning has been the subject of increasing interest over the last decades, due to its clinical implications for early neurobehavioral assessment [1–3] as well as for its theoretical insights into the ontogenetic origins of a wide arrays of phenomena, including auto-regulation [4, 5], mirror-like behaviors [6], interoception and arousal [7], consciousness [8] and communication [9].

Being able to identify fetal yawns could serve various potential clinical interests. In fact, it has been proposed that increased rates of yawning might help to identify high risk fetuses [2, 3], while lack of fetal yawn can be predictive of brainstem dysfunction after birth [7].

**Competing interests:** The authors have declared that no competing interests exist.

Therefore, having an accurate, reliable method for coding fetal yawns is crucial not only for the study of the development of perinatal behavior, but also for allowing research results to be implemented in actual clinical practices.

However, the intrauterine development of yawning is still poorly understood, as indicated by inconsistencies among published studies in the estimates of yawning frequencies within the same windows for gestational age (GA). In particular, as shown in Table 1, during the third trimester of pregnancy, the average number of yawns observed per hour varied from zero [10] to 14 [11]. The sharp differences between these results might be partially explained by different factors, including fetal circadian rhythms and pathological conditions [2, 3]. However, all studies included healthy fetuses [3, 10–19] and most had US scans performed during the afternoon [11, 12, 13, 14, 15, 18, 19]. Therefore, the inconsistencies shown in Table 1 suggest that the measures used in these studies lacked adequate reliability or validity.

Most studies adopted qualitative criteria in identifying yawns based on the de Vries et al. definition (Table 2). An alternative approach defined yawns in a more operational and quantitative way as those mouth openings where the time to maximum opening was longer than the time from maximum opening to closure [10].

## Reliability and validity issues

In order to overcome the observed inconsistencies in coding yawns from ultrasound scans, we need to address two distinct issues: the precision or inter-rater reliability of measures used to code yawns and the accuracy or construct validity of the measures, i.e., whether the measures used can discriminate yawning from other fetal behaviors involving mouth opening, such as swallowing, mouthing or distress expressions.

Kanenishi et al. [18] and Sato et al. [19] mentioned the reliability issue in relation to discrepancies between their findings and those from other studies. They concluded that the subjectivity of methods used to identify yawns and other behaviors from US scans might have resulted in low inter-rater agreement. To the best of our knowledge, however, only the Reissland et al. [10] study included two independent coders. In that study, calculation of Cohen's Kappas showed good reliability. On the other hand, the issue of construct validity, i.e., the accuracy of identifying yawns and distinguishing them from other actions involving mouth opening, has not been addressed in any published study.

The criterion proposed by Reissland et al. [10] is the only formalized, quantitative system for assessing fetal yawns. It can therefore be reproduced and tested for both reliability and validity. Moreover, since it relies only on temporal cues pertaining to mouth movements, it has the advantage of being particularly simple and easy to apply, especially when dealing with US scans, often characterized by limited spatio-temporal resolution, partial accessibility of the face to observation, and imaging artifacts [20].

However, this simplification might carry the risk of sacrificing accuracy, because the authors did not address the crucial construct validity issue of whether the system can distinguish yawning from other movements involving mouth opening. Reissland et al. [10] justified their methodological choices by citing Petrikovski et al. [3], who reported that the opening phase of yawns is longer than the closing phase, but they did not offer any evidence for the complementary assumption on which the system is based, i.e. that yawns are the *only* mouth opening actions that meet this criterion. Consequently, this method could be prone to type I errors (false positives).

## Neonatal yawns as a model for coding fetal yawns

One undisputed feature of yawning is the apparent stability of its behavioral pattern throughout life [21, 22]. Fetal yawns, in particular, have been reported to be similar to the behavioral

**Table 1. Spontaneous yawning frequencies of healthy fetuses.**

| Reference | US Technique | N | GA range (w) | Duration (h:min) | Yawns/h (SD) |
|---|---|---|---|---|---|
| Van Woerden et al. (1988) [12] | 2D | 19 | 38–40 | 01:00 | 4.3 (NA) |
| Petrikovsky et al. (1999) [3] | 2D | 16 | 36–40 | 01:00 | 5.0 (4.0) |
| Kurjak et al. (2003) [13] | 4D | 10 | 30–33 | 00:15 | 4.0 (NA)° |
| Kurjak et al. (2004) [11] | 4D | 10 | 33–35 | 00:15 | 14.0 (NA)° |
| Reissland et al. (2012) [10] | 4D | 14 | 24 | 00:10 | 11.6 (13.0) |
| Reissland et al. (2012) [10] | 4D | 15 | 28 | 00:10 | 8.4 (12.2) |
| Reissland et al. (2012) [10] | 4D | 15 | 32 | 00:10 | 4.4 (5.8) |
| Reissland et al. (2012) [10] | 4D | 14 | 36 | 00:10 | 0.0 (0.0) |
| Yan et al. (2006) [14] | 4D | 10 | 28–34 | 00:15 | 11.6 (7.2) |
| Kanenishi et al. (2013) [18] | 4D | 24 | 25–28 | 00:15 | 4.5 (4.3) |
| Sato et al. (2014) [19] | 4D | 23 | 20–24 | 00:15 | 2.6 (3.5) |
| Yigiter and Kavak (2006) [16] | 4D | 63 | 11–40 | 00:30 | 7.0 (3.5) |
| AboEllail et al. (2018) [15] | 4D | 25 | 30–31 | 00:15 | 0.0 (NA)° |
| AboEllail et al. (2018) [15] | 4D | 43 | 32–35 | 00:15 | 0.0 (NA)° |
| AboEllail et al. (2018) [15] | 4D | 43 | 36–40 | 00:15 | 4.0 (NA)° |

Note: Values in parentheses represent Standard Deviations; N = Sample Size; Duration = Duration of a single observation session;° values calculated based on reported medians; NA = Not Available.

pattern exhibited after birth by neonates and infants [12, 17, 23, 24]. Therefore, the analysis of yawns in neonates represents a good model for testing and developing methods for coding yawns in fetuses, especially because the superior spatio-temporal resolution and clarity of video-recordings allow a more accurate and precise identification of yawns. The population of healthy preterm neonates, in particular, seems to be the best suited for this purpose, since they are the closest to fetuses with regard to developmental level.

**Table 2. Operational definitions of fetal yawning.**

| Reference | Definition |
|---|---|
| Van Woerden et al. (1988) [12] | *Similar to the yawn observed after birth; a prolonged wide opening of the mouth followed by a quick closure, and mostly combined with a retroflexion of the head* (p. 99) |
| Petrikovsky et al. (1999) [3] | *Yawning was defined as a prolonged wide opening of the mouth followed by a quicker closure of the mouth* (pp. 127,128) |
| Kurjak et al. (2003) [13] | *Slow and prolonged wide opening of the jaws followed by quick closure with simultaneous retroflexion of the head and sometimes elevation of the arms of exoration* (p. 499) |
| Yan et al. (2006) [14] | *Yawning [was defined] as a slow, wide, prolonged opening of the jaws followed by quick closure with simultaneous retroflexion of the head* (p. 110) |
| Reissland et al. (2012) [10] | *We defined a yawning event to be those mouth openings where the time to maximum opening of the mouth was of a longer duration than the time from maximum opening to closing* (p. 3) |
| Kanenishi et al. (2013) [18] | Video Sample |
| Yigiter and Kavak (2006) [16] | *Slow and prolonged wide opening of the jaws followed by quick closure with simultaneous retroflexion of the head and sometimes elevation of the arms of exoration* (p. 708) |
| Sato et al. (2014) [19] | Video Sample |
| AboEllail et al. (2018) [15] | *Yawning represented prolonged wide and slow jaw opening followed by quick closure with simultaneous head retroflexion* (p. 2) |

### Research strategy

The aim of Study 1 was to assess the construct validity of the criterion for coding yawns proposed by Reissland et al. [10]. We first identified mouth openings in videos of preterm infants adopting the same Baby FACS-based criteria used to identify mouth opening in their study, and then, using the same timing-based criterion they proposed, we categorized mouth openings as yawns or non-yawns.

We then tested the agreement between yawns identified according to their criterion and according to a behavioral description, contained in the System for Coding Perinatal Behavior (SCPB) [25]. SCPB is a coding system that focuses on complex motor patterns occurring in the face, head, and upper trunk adapted from Oster's Facial Action Coding System for Infants and Young Children (Baby FACS) [22], a comprehensive, anatomically based coding system with established reliability and validity and earlier studies in the literature. Baby FACS Action Units (AUs) represent discrete, minimally distinguishable actions of the facial muscles, allowing any facial movement to be precisely and objectively identified in terms of combinations and sequences of its constituent facial muscle actions.

In Study 2 we developed a new machine learning system for identifying yawns based on temporal dynamics of mouth openings and tested its validity on the same sample of videos analyzed in Study 1. We adopted the description from the SCPB to train and test support vector machine (SVM) algorithms. In order to enhance replicability, the methods for both studies are available on protocols.io at 10.17504/protocols.io.739hqr6.

## Methods—Study 1

### Participants

Seventeen preterm neonates admitted to the Neonatal Intensive Care Unit (NICU) at SS. Annunziata Hospital of Cosenza (Italy) participated in the study upon informed consent from parents.

The study sample included 17 healthy preterm neonates (5 males and 12 females) born between 26 and 33 weeks GA (M = 29.61; SD = 2.08), with birth weight appropriate for gestational age (AGA) observed between 28 and 35 weeks PMA (M = 31.41; SD = 1.98). Exclusion criteria were: congenital anomalies, heart or metabolic disorders, fetal infections, clear teratogenic factors, Apgar at five minutes < 6 and grade III or IV hemorrhages.

### Procedure

Neonates were observed while they were lying supine in a cot. Behavior was video-recorded (24 frames per second) for 10 to 30 minutes (M = 18.63, SD = 6.31), at a midpoint in the feeding cycle, when the neonates were not receiving any stimulation through routine nursing or medical care.

### Coding methods

**Identification of mouth opening.** Reissland et al. [10] identified mouth opening as movements in which the mouth was stretched widely open and the mandible was pulled down vertically. This definition is based on the appearance changes for AU 27 described in FACS [26] and Baby FACS [22], as illustrated in Reissland et al. [10]. For this reason, we coded mouth opening in the preterm neonates every time the lips parted (AU 25) and the mouth stretched widely open (AU 27) simultaneously [22]. Frame by frame coding of the video-recordings was performed by two independent coders expert in FACS, Baby FACS, and micro-analysis of neonatal behavior. The secondary coder did reliability coding for 41% of the video recordings.

Videos were coded using ELAN, professional software for the creation and management of complex annotations on video and audio (Max Planck Institute for Psycholinguistics, The Language Archive, Nijmegen, The Netherlands; http://tla.mpi.nl/tools/tla-tools/elan/).

**Identification of yawns based on timing of mouth opening and closing (Criterion A).** In order to replicate the Reissland et al. [10] procedure for identifying mouth openings as yawns, coders scored two different time intervals for each mouth opening:

- total duration was coded from the onset of mouth opening, the first frame where the mouth opening motion was visible, to the offset, the last frame where mouth opening was visible.

- plateau duration was coded from the first frame (plateau onset) to the last frame (plateau offset) during which maximum mouth opening was maintained.

Mouth openings were subsequently categorized as yawns or not yawns based on whether or not they satisfied the Reissland et al. [10] criterion defining yawns as mouth openings in which "the time to maximum opening of the mouth was of longer duration than the time from maximum opening to closing" [10; p. 3].

Consistent with the definition provided in the Fetal Observable Movement System [27; FOMS], the plateau (the portion of the episode where mouth opening remained at its apex), albeit scored separately from the opening and closing phases, was considered "to be part of the opening rather than the closing phase" (p. 169), and the closing phase was timed from the end of the plateau.

**Identification of yawns based on SCPB (Criterion B).** Two independent expert FACS and Baby FACS coders identified mouth openings as yawns according to the following definition from *The System for Coding Perinatal Behavior* (SCPB) [25] based on the AUs described in the comprehensive, anatomically based Facial Action Coding System for Infants and Young Children (Baby FACS) [22] and previous studies in the literature [17, 21].

Yawning (AU 94) is a stereotyped behavior characterized by a slow mouth opening with deep inspiration, followed by a brief apnea and a short expiration and mouth closing. One of the characteristic features of yawning is its timing, with a gradual acceleration followed by an abrupt deceleration of the facial actions involved. Yawning usually emerges from a relaxed face, initially involving mouth stretching widely open (AUs 25 + 27) and upper eyelids drooping (AU 43). Although the specific AUs accompanying yawns vary, at apex they may include tightly closed eyelids (AUs 6+7+43), flattened tongue shape (AU 76b), and swallowing (AU 80). During the plateau, brow knitting (AU 3), brow knotting (AU 4), nose wrinkling (AU 9), lateral lip stretching (AU 20), nostril dilatation (AU 38) and head tilting back (AU 53) may occur. In this phase, the expansion of the pharynx can quadruple its diameter, while the larynx opens up with maximal abduction of the vocal cords [21]. Yawning is often accompanied by limb stretching [17] and other bodily movements.

The SCPB [25] is a recently developed coding scheme based on frame-by-frame analysis of video-recorded material. The system focuses on fetuses, preterm and full-term neonates and aims to identify and reliably code the repertoire of complex patterns of behaviors observable from the last trimester of pregnancy to the first month of age. The SCPB is a system based on the Action Units (AUs), Action Descriptors (ADs), Miscellaneous Actions and Supplementary Codes described in Baby FACS [22] for identifying discrete facial muscle actions and more complex, configurationally defined patterns of facial behaviors.

## Inter-rater reliability

Cohen's Kappa was used to assess inter-rater reliability between the primary and secondary coders. Reliability was separately assessed for identifying mouth opening and for identifying

yawns. The assessment of inter-rater reliability for identifying mouth opening, using a time window of one second for both onset and offset, resulted in an acceptable agreement between coders (kappa = .72).

Inter-rater reliability in identifying yawns was then assessed for Criterion A and criterion B. Acceptable reliability was obtained using Criterion A (kappa = .64), while perfect inter-rater agreement was found using criterion B (kappa = 1).

### Data analysis: Distinguishing yawns from non-yawn mouth openings

Each mouth opening episode was categorized dichotomously as a yawn (1) or non-yawn mouth opening (0) according to Criterion A and Criterion B. In order to assess the accuracy of the measure adopted by Reissland et al. [10], we used a contingency table to assess its sensitivity (i.e. the true positive rate, calculated as the proportion of yawns that were correctly classified as yawns) and specificity (i.e. the true negative rate, calculated as the proportion of non-yawns that were correctly classified as non-yawns), and compute Cohen's Kappa.

## Results—Study 1

Over the 17 video-recordings, 130 mouth opening episodes were scored. The average duration of mouth openings was 2.48 s (SD = 1.41), and the average duration of the plateau was 0.63 s (SD = 0.69). Eighty-eight mouth openings (67.7% of total episodes) were classified as yawns according to criterion A [10], while only 15 (11.5% of total episodes) were recognized as yawns according to criterion B.

All 15 of the yawns identified according to criterion B also satisfied the temporal criterion specified by Reissland et al. [10] (criterion A). However, 73 mouth openings identified as yawns by criterion A were not coded as yawns according to Baby FACS [22], the SCPB [25], and earlier behavioral descriptions in the literature (criterion B). The remaining 42 episodes were classified as non-yawn mouth openings according to both criteria. As a consequence, despite a perfect sensitivity (1.00), the estimated specificity for Criterion A was 0.36, resulting in a reliability barely above chance (Cohen's Kappa = 0.12).

## Discussion—Study 1

The results of Study 1 revealed that the Reissland et al. [10] criterion for classifying yawns (Criterion A) had low specificity, producing a high rate of false positives (73 of 130 mouth openings, 56% of total episodes). The criterion for yawning proposed by these researchers was satisfied by 88 (67.7%) of the 130 episodes of mouth opening in our sample, excluding only mouth openings where the closing phase was longer than the sum of the mouth opening and plateau phases. It is notable that the high rate of yawns coded by criterion A is consistent with the results reported in the original study by Reissland et al. [10], who classified 56 out of the 83 (67.5%) mouth openings they observed in fetuses as yawns. The performance of Criterion A reflects the limited power of the criterion based on only two temporal landmarks to distinguish yawning from non-yawn mouth opening. In sum, it is clear that we cannot consider the overly simple quantitative criterion for identifying yawns proposed by Reissland et al. [10] to be an accurate, valid method for distinguishing between yawns and non-yawn mouth openings in fetuses or preterm neonates.

Despite these results, it should be possible to develop a quantitative criterion for identifying yawns based just on the temporal dynamics of mouth opening and closing. The perfect sensitivity exhibited by the Reissland et al. [10] criterion confirms that yawns are characterized by distinctive temporal dynamics. However, their criterion overlooked additional temporal variables that could be relevant for identifying yawns. In Study 2 we examined the potential

usefulness of several derived measures for distinguishing yawning from non-yawn mouth opening in addition to those analyzed in Study 1.

## Methods—Study 2

In Study 2 we conducted a more detailed analysis of the temporal dynamics of yawning to assess the feasibility of distinguishing yawning from non-yawn mouth opening based solely on temporal cues coded from videos or US scans of mouth and jaw movements.

Following a first exploratory phase, we adopted a support vector machine (SVM) approach that was cross-validated on the same sample of preterm neonates used in Study 1. SVMs are supervised learning models with associated learning algorithm that analyze data used for classification and regression analysis, supporting high dimensional data. These methods are widely used in different research fields such as bioinformatics, text mining, face recognition and image processing and are regarded, along with neural networks and fuzzy systems [28], as a state-of-the-art tool for machine learning. SVM-based systems have also been used for automated facial behavior analysis in some pioneering studies [29–33]. Participants and settings were the same as described for Study 1.

### Coding method

In addition to coding the total duration of mouth opening and duration of the plateau, as defined in Study 1, we calculated the following variables:

- Duration of the opening phase: from mouth opening onset to plateau onset
- Duration of the closing phase: from plateau offset to mouth opening offset
- opening/closing asymmetry: difference between the durations of the opening and closing phases
- opening/closing ratio: ratio of the duration of the opening phase to the duration of the closing phase

### Reliability

In order to establish the reliability of these measures, we compared the duration of the three phases (opening, plateau and closing) as scored by the two independent coders for Study 1. The differences between the two coders were deemed acceptable for the duration of each of the three phases: opening (Median = .08 s), plateau (Median = .10 s) and closing (Median = .12 s). Adopting a tolerance window of .5 seconds, the percentage agreement between coders was 93% (39 out of 42) for both opening and plateau duration and 95% (40 out of 42) for closing duration.

### Hierarchical logistic regressions

Preliminary analyses were carried out, via hierarchical linear regression models, in order to identify specific features of yawns compared to other mouth openings in terms of their temporal dynamics, and to investigate the relations between different parameters for the two classes of episodes.

### Machine learning algorithm

Based on findings from the exploratory analysis, the use of Support Vector Machine (SVM) classifiers was deemed appropriate to maximize the classification margin and minimize the

risk of type I (false positives) as well as type II errors (false negatives) in distinguishing yawns and non-yawn mouth openings. We used LibSVM [34] with radial basis functions (RBF) kernel [32] in R, version 3.5.2 (package "e1071"), to build our models and to generate predictions for our test cases. Grid search based on 10-fold cross-validation error was employed to optimize the parameters C and gamma using the function "tune.svm", within the interval [$10^{-3}$, $10^3$]. Model A was optimized with C = 100 and gamma = 0.1 while Model B was optimized with C = 10 and gamma = 1.

Classification performance was calculated via percent agreement and Cohen's Kappa calculation for two different SVM models by changing the subset of episodes used respectively for training and testing, in order to identify and cross-validate the optimized model. Since yawns are expected to share distinctive temporal dynamics, in addition to the full model including main effects and interactions of all variables (total duration, plateau duration, duration of the opening phase, opening/closing asymmetry and opening/closing ratio; Model A), a second model was tested including only the three-ways interaction effects of total duration, plateau duration and opening/closing asymmetry (Model B).

We compared the classification performance of the two SVM models via hold-out validation. On each iteration, two thirds (n = 87) of the sample (N = 130) were randomly assigned to training, while the remaining 43 episodes were used for testing. For each model, we computed overall agreement, Cohen's Kappa, sensitivity (true positive rate) and specificity (true negative rate). The candidate model with the highest Cohen's Kappa coefficient was deemed the best model for yawn identification.

## Results—Study 2

### Hierarchical logistic regressions

Hierarchical linear regressions highlighted several differences between yawns and non-yawn mouth openings involving the study variables (see Table 3 for descriptive statistics). In particular, total duration was longer for yawns (Mean = 5.123, SD = 1.252) compared to non-yawn mouth openings (Mean = 2.14, SD = 1.01), t(126) = 9.93, $p < .001$, β = .652. Moreover, yawns had a longer opening phase (Mean = 2.01, SD = 0.65) than non-yawn mouth openings (Mean = 0.77, SD = 0.56), t(126) = 7.66, $p < .001$, β = .561. Plateau was also longer for yawns (Mean = 2.10, SD = 0.83) compared to other mouth openings (Mean = 0.44, SD = 0.36), t(126) = 13.88, $p < .001$, β = .779. The duration of the closing phase, however, did not differ significantly between the two classes of behavior, t(126) = -.35, $p = .726$, β = -.054 (see Fig 1). As expected, yawns also presented a stronger opening/closing asymmetry than non-yawn

**Table 3. Variables related to temporal dynamics of yawns and non-yawn mouth openings.**

| Variable | Yawns (n = 15) | | | | Non-Yawn Mouth Openings (n = 115) | | | |
|---|---|---|---|---|---|---|---|---|
| | Mean (SD) | Median | Min | Max | Mean (SD) | Median | Min | Max |
| Duration | 5.12 (1.25) | 5.23 | 3.00 | 6.97 | 2.14 (1.01) | 1.92 | 0.71 | 5.95 |
| Plateau Duration | 2.10 (0.83) | 2.24 | 1.04 | 3.20 | 0.44 (0.36) | 0.36 | 0.04 | 1.88 |
| Opening Duration | 2.01 (0.65) | 1.98 | 1.12 | 3.67 | 0.77 (0.56) | 0.59 | 0.13 | 3.84 |
| Closing Duration | 1.01 (0.36) | 0.96 | 0.53 | 2.01 | 0.92 (0.58) | 0.81 | 0.11 | 3.39 |
| Asymmetry | 1.00 (0.43) | 0.92 | 0.39 | 1.66 | -0.15 (0.70) | -0.15 | -2.64 | 2.33 |
| Opening/Closing Rate | 2.06 (0.53) | 1.99 | 1.38 | 3.42 | 1.11 (0.96) | 0.78 | 0.11 | 4.94 |

Note: Values are expressed in seconds

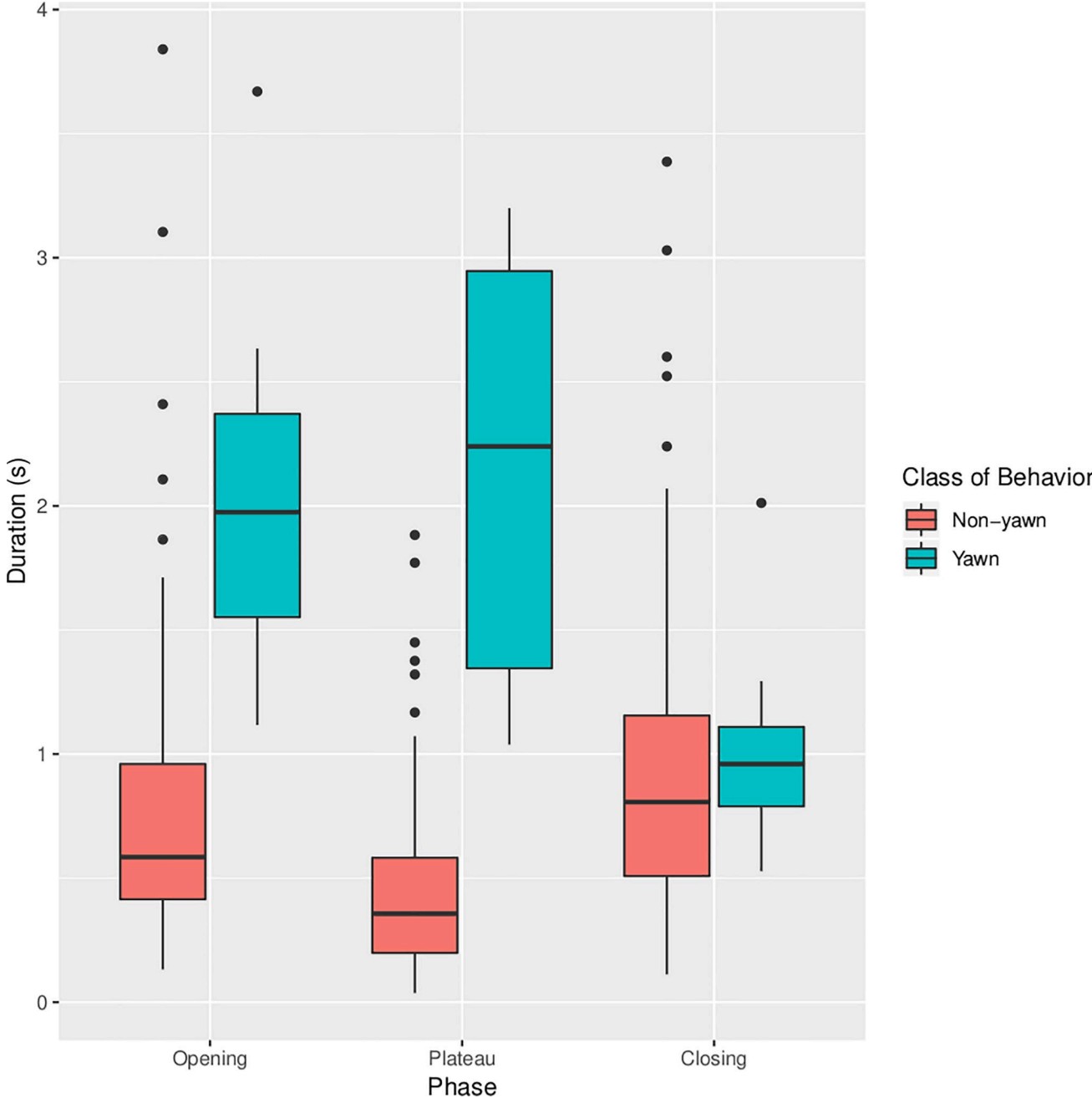

**Fig 1. Temporal dynamics of yawning and non-yawn mouth openings.** Box plot of the duration of the opening, plateau and closing phases of yawn and non-yawn mouth openings. The lower and the upper hinges represent, respectively the first and third quartiles; the whiskers extend from the hinges to the most extreme value no further than 1.5* from the hinge. Points represent outliers.

mouth openings, $t(126) = 6.34$, $p < .001$, $\beta = .496$, as well as a larger opening/closing ratio, $t(126) = 4.12$, $p < .001$, $\beta = .349$.

## SVM models evaluation

The hold-out validation procedure revealed overall good performance, with both models showing an agreement above 98% with the SCPB-based classification. In particular, the full

model (Model A) highlighted the best classification performance, showing almost perfect specificity (M = 1.00, SD = 0.01) and good sensitivity (M = 0.93, SD = 0.10). Having achieved the highest Cohen's Kappa coefficient (M = 0.94, SD = 0.08), Model A was deemed the best model for classifying yawns and non-yawn mouth openings. Model B, defining the probability of a mouth opening to be a yawn as a function of the interaction of total duration, plateau duration and opening/closing asymmetry, performed only marginally worse in terms of Cohen's Kappa (M = 0.92, SD = 0.12), and even outperformed Model A in terms of sensitivity (M = 0.95, SD = 0.11), while also displaying good specificity (M = 0.99, SD = 0.02).

## General discussion

The ability to identify yawns in fetuses and to distinguish yawns from non-yawn mouth openings is of interest because accurate detection of this widely observed behavior can have potential clinical importance for identifying early signs of neurodevelopmental abnormalities and other conditions such as placental diseases [35] in fetuses, very early preterm infants, and other populations at risk.

We reported two studies that examined the temporal dynamics of yawning in preterm neonates comparable in gestational age to fetuses observed in previously reported ultrasound studies of fetal yawning [2, 3, 10–16, 18, 19]. In Study 1 we tested the reliability and construct validity of the only quantitative measure for identifying fetal yawns in the literature by applying the same time-based criterion to yawns showed by preterm neonates coded with a more detailed behavioral coding system (The *System for Coding Perinatal Behavior*, SCPB) [25] adapted from the comprehensive, anatomically based Facial Action Coding System for Infants and Young Children (Baby FACS) [22]. The results of Study 1 revealed that the Reissland et al. [10] criterion for distinguishing between yawns and non-yawn mouth opening had low specificity, producing a high rate of false positives.

In Study 2, we developed and evaluated two Support Vector Machine (SVM) models for classifying perinatal yawns and non-yawn mouth openings. Our results demonstrate the potential of SVM for identifying yawns based on a quantitative analysis of the temporal dynamics of mouth openings. In particular, the fact that the partial model (Model B), only including a three-way interaction effect of total duration, plateau duration and opening/closing asymmetry (defined as the difference between opening and closing durations) performed only marginally worse than the full model is consistent with previous knowledge. In fact, yawning is known to be a stereotyped behavioral pattern characterized by a prolonged mouth opening, including an extended plateau and a short closing phase [3, 12–14, 22], and this description can be formalized as an interaction of the three variables best representing these features. Moreover, the fact that the episodes coded as yawns occupy a specific region of the three-dimensional space defined by duration, plateau duration, and opening/closing asymmetry is consistent with the prevalent view of yawning as a distinctive type of mouth opening, therefore indirectly confirming the accuracy and precision of the identification criteria based on SCPB that we adopted (Criterion B). Further research is needed in order to confirm these results on bigger samples, as well as to provide additional training for the SVM algorithm.

The proposed machine-learning system might represent a crucial asset for the study of fetal yawning, making it possible to distinguish between yawns and other behavioral patterns involving mouth opening, e.g. swallowing, mouthing, or distress expressions. In fact, it represents the first cross-validated and easily reproducible method for coding yawns at early developmental stages, and, because it is based on the temporal analysis of mouth openings, it can be used with 4D US scans. In particular, this method–developed using preterm neonates as a training population–enables us to overcome the interobserver reliability limitations of

descriptive approaches to yawn recognition, while also being able to avoid the risk of construct validity issues resulting from the adoption of an oversimplified criterion like the one proposed by Reissland et al. [10].

In order to establish this method as a viable option for identifying fetal yawns in clinical and research settings, additional work should be done to test it on fetal behavior. In particular, the suboptimal framing and variable quality that characterize ultrasonographic scans represent a ubiquitous issue in the study of fetal behavior. However, the fact that the proposed method only relies on the timing of mouth openings highlights a key advantage of this approach, that was specifically conceived in order to overcome the issues of identifying a complex facial behavior such as yawn from ultrasonographic scans. Another potential reason for concern regards the different conditions of fetuses compared with preterm neonates; for example, fetuses are immersed in amniotic fluid and the yawn of the near-term fetus might occasionally be mechanically constrained because of its position. However, there is unanimous agreement in the literature on the stability of the yawning behavioral pattern throughout life [12, 17, 21, 22, 23, 24]. Therefore, the advantages of using preterm neonates as a model for training classificatory algorithms far outweigh these potential issues.

A strength of the current study was the use of the SCPB [25], a rigorous coding system based on Baby FACS [22], to identify yawns. Because Baby FACS coding is based on multiple and redundant cues to each facial action, facial muscle actions can be reliably identified in fetuses [36], preterm and full-term neonates [37], and infants with facial anomalies [38] as well as typically developing infants. And because the basic coding units of Baby FACS are exhaustive and mutually exclusive, any complex facial movement can be precisely and unambiguously identified in terms of combinations and sequences of its constituent facial muscle actions. Therefore, Baby FACS, which has been referred to as the "gold standard" for coding infants' facial expressions [39], is especially suited for coding yawning, sensory and perceptual responses [37], pain [40], and other fetal, neonatal and infant behaviors that don't fit simplified templates for a limited set of emotional expressions.

The reliability of the proposed approach for fetal yawning classification is conditional on a preliminary evaluation of the temporal resolution of the 4D US scans used and of the resulting errors in assessing the duration of the parameters of interest. However, the recent development of ultrasonographic machines performing up to 25 frames per second [14], together with the specificities highlighted by yawns with regard to temporal dynamics are encouraging for future applications.

Because coding micro-analytically mouth openings and their plateaus is time-consuming, the utility of this system for coding neonatal yawns would be greatly improved by the implementation of a machine-controlled method for tracking neonatal mouth openings, which would make the identification process fully automated. This accomplishment would guarantee a quick, valid and reliable system for investigating yawns in fetuses and neonates.

In conclusion, the development of a reliable and valid method for identifying yawning can provide a potentially valuable tool for studying perinatal behavior and for assessing fetal and preterm infant wellbeing. Future research using machine-learning systems could contribute to the development of sensitive measures for early diagnosis of neurological impairments and other disorders.

## Supporting information

**S1 File. Dataset.**
(CSV)

## Acknowledgments

We are deeply indebted to Gianfranco Scarpelli and the nursing staff at the Neonatal Intensive Care Unit (NICU) of the SS. Annunziata Hospital of Cosenza for their invaluable cooperation. We also wish to thank Mafalda Marinaro for technical support and Tiziana Aureli for commenting an earlier version of these studies, both included in the Ph.D. dissertation of the first author.

## Author Contributions

**Conceptualization:** Damiano Menin, Marco Dondi.

**Data curation:** Damiano Menin, Angela Costabile, Flaviana Tenuta.

**Formal analysis:** Damiano Menin, Marco Dondi.

**Investigation:** Angela Costabile, Flaviana Tenuta.

**Methodology:** Damiano Menin, Angela Costabile, Harriet Oster, Marco Dondi.

**Project administration:** Marco Dondi.

**Resources:** Angela Costabile, Flaviana Tenuta, Marco Dondi.

**Software:** Damiano Menin.

**Supervision:** Harriet Oster, Marco Dondi.

**Validation:** Damiano Menin.

**Visualization:** Damiano Menin.

**Writing – original draft:** Damiano Menin, Marco Dondi.

**Writing – review & editing:** Damiano Menin, Angela Costabile, Flaviana Tenuta, Harriet Oster, Marco Dondi.

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
