## [Decision Letter · Decision Letter 0]

24 Sep 2019

PONE-D-19-18762

Identifying fetal yawns based on temporal dynamics of mouth openings: A preterm model using support vector machines (SVMs)

PLOS ONE

Dear Prof. Dondi,

Thank you for submitting your manuscript to PLOS ONE. After careful consideration, we feel that it has merit but does not fully meet PLOS ONE’s publication criteria as it currently stands. Therefore, we invite you to submit a revised version of the manuscript that addresses the points raised during the review process.

The manuscript and the reviewers’ comments were carefully evaluated. The manuscript was appreciated by the Reviewers. Nevertheless, as suggested, the manuscript requires some improvement before to be considered for publication. Further suggested revisions are in detail reported in the Reviewers’ comments.

We would appreciate receiving your revised manuscript by Nov 08 2019 11:59PM. To enhance the reproducibility of your results, we recommend that if applicable you deposit your laboratory protocols in protocols.io, where a protocol can be assigned its own identifier (DOI) such that it can be cited independently in the future. For instructions see: http://journals.plos.org/plosone/s/submission-guidelines#loc-laboratory-protocols

We look forward to receiving your revised manuscript.

Kind regards,

Simone Garzon

Academic Editor

PLOS ONE

Journal Requirements:

Additional Editor Comments (if provided):

Reviewers' comments:

Reviewer's Responses to Questions

**Comments to the Author**

1. Is the manuscript technically sound, and do the data support the conclusions?

Reviewer #1: Yes

Reviewer #2: Yes

Reviewer #3: Yes

Reviewer #4: Yes

2. Has the statistical analysis been performed appropriately and rigorously? 

Reviewer #1: Yes

Reviewer #2: I Don't Know

Reviewer #3: No

Reviewer #4: No

3. Have the authors made all data underlying the findings in their manuscript fully available?

Reviewer #1: Yes

Reviewer #2: No

Reviewer #3: No

Reviewer #4: Yes

4. Is the manuscript presented in an intelligible fashion and written in standard English?

Reviewer #1: Yes

Reviewer #2: Yes

Reviewer #3: Yes

Reviewer #4: Yes

5. Review Comments to the Author

Reviewer #1: I have no serious objection to any part of the paper. The conceptual design of the paper and its intent are clear. The observation of fetal movement in general is important to clinicians as well as those that study fetal development. This paper makes a solid contribution to this field of study.

The authors assume that the preterm neonate is a valid model for the fetus. My personal experience supports this assumption and I agree that it is certainly the best way to approach the problem.

The authors demonstrate the need for an improved definition of neonatal yawning. They define quantifiable features that can be measured during neonatal yawning that can be used to train a support vector machine (SMV). Anyone should be able to duplicate the work to the extent that a given set of data and a SMV with allow. In Fig 1, the degree to which the distributions are separated suggest that any number of machine learning techniques would be successful. Therefore, I'm satisfied with the sample size and methods they have used.

I do have concerns:

The neonates in this study were placed in a convenient position for observation during data collection, but the fetus will not be so co-operative. Moreover, the yawn of the near term fetus might occasionally be mechanically constrained because of its position. Further, the neonate is yawning in air, while the fetus will be immersed in amniotic fluid. The timings and definitions may need to be adjusted.

The paper would benefit from more data, sufficient to create an independent test set of equal size. The observation that your Model B is the best performing may be a statistical artifact, or a bit of over training, or it may be real. I can't say.

I wish the authors well in their future work.

Reviewer #2: The article deals with the problem of detecting yawning in fetuses (preterm neonates) based on temporal dynamics of mouth openings. The authors conducted two studies. In the first, the accuracy of detection based on the method from the literature [10] was compared with the behavioral description (SCPB). In the second study the machine learning technique was applied: five SVM models were considered. According to the Authors, the second approach (SVM) is better.

The paper represents good quality, detailed comments are given below:

1) In the title it is probably better to replace “preterm model” with “preliminary model” or “initial model”. Unless the adjective “preterm” refers to neonates.

2) How were SVM parameter values determined (C=100, gamma=1, line 316) and what does "type 1 error" and "type 2 error" (lines 313-314) mean?

3) It would be worth describing more precisely how sensitivity and specificity were determined in study 1.

4) The abbreviation "AU" is explained later (line 190) than its first use in the text (line 156).

5) What does "t(126)" (lines 346-353) mean?

6) English can be slightly improved in some places, e.g.:

- “construct validity of the measure” -> “validity of the measure” or “correctness of the measure”

- “risk of sacrificing accuracy for precision” -> “risk of lowering accuracy at the expense of precision” (line 96)

- line 385: “having achieved the highest” -> “having the highest” or “achieving the highest”

- line 389: “did not just display a longer” -> “not only showed a longer”

- line 394: “according to one specific iteration” -> “in a given iteration”

- line 156: “et al” -> “et al.”, “F For” -> “For”, line 383: “011” -> “0.11”

The above minor comments do not affect the generally high quality of the article. In my opinion, after these minor corrections, the article should be published in the PLOS ONE Journal.

Reviewer #3: The paper concerns the problem of identifying and modeling of fetal yawns from ultrasonographic scans by analyzing temporal dynamics of fetal mouth openings. The results of two main studies were presented. In the first, the validity of the criterion for coding fetal yawns was investigated while in the second, the SVM based classifiers to distinguish the perinatal yawns from non-yawn mouth openings were investigated. Five different SVM models were proposed and the presented results shown high efficiency of RBF based kernel SVM of fetal yawns recognition.

The addressed issues are interesting and provides good insight into the problems of constructing the automated systems to support the early assessment of fetal neurobehavioral development. The work is written carefully, with clearly defined objective. The following are general and more specific comments

1) The abbreviation usage needs improving. Each abbreviation shall be explained at the first use. The “symbol” t(126) (page 10) needs explanation as well?

2) There are some typos, e.g.: “F For” -> “For” (line 156), “011” -> “0.11” (line 383). The text should be checked thoroughly.

3) How were SVM parameter values (C and γ) determined?

4) Is the classification performance defined on the testing sets only? If yes, the testing set would contain 5 yawing cases (on average). As the generalization ability estimation is of the crucial importance, is this number enough for making the conclusions about fetal yawn models?

5) It seems that the figure 2 shows the SVM classification result for Model A (which is not explicitly stated). But what is its purpose? Why plateau duration was fixed at 1.05 s. (half of the mean of plateau duration related to fetal yawns)?

6) As the classification results are so high, did Authors consider to model the fetal yawns by using only two temporal dynamics variables of the highest discrimination ability (like opening and plateau phases – see Figure 1)?

Considering the above, my recommendation is to accept the paper after these minor corrections and comments are addressed.

Reviewer #4: I read with great interest the Manuscript titled “Identifying fetal yawns based on temporal dynamics of mouth openings: A preterm model using support vector machines (SVMs)” (PONE-D-19-18762), which falls within the aim of PLOS ONE. In my honest opinion, the topic is interesting enough to attract the readers’ attention. The manuscript is well written, methodology is accurate, and conclusions are supported by the data analysis. Nevertheless, authors should clarify some points and improve the discussion citing relevant and novel key articles about the topic.

Authors should consider the following minor recommendations:

- Introduction. Regarding the difference among reported yawns frequencies, which is the role of circadian changes in yawns activity of the fetus. Does yawns activity change over time? I would suggest discussing this point.

- The Authors did not mention the sample size calculation for their study. I would suggest reporting how the Authors chosen the number of included neonates.

- The authors have not adequately highlighted the strengths and limitations of their study. I suggest better specifying these points. The Authors use the Baby FACS coders as the gold standard for the evaluation of yawns, is it actually the gold standard? – The newborns yawns in air and fetus yawns in water, do Authors expect some changes?

- As appropriately stated by the authors, fetal yawning has been associated with proper neurological development. In this regard, the possibility to detect these fetal movements may play a pivotal importance as additional tool to assess fetal wellbeing in utero. I would suggest stressing the possible underlying causes of fetal neurological impairment, such as for example placental diseases (refer to: PMID: 28282763; PMID: 28466013; PMID: 28243732).

6. PLOS authors have the option to publish the peer review history of their article (what does this mean?). If published, this will include your full peer review and any attached files.

Reviewer #1: No

Reviewer #2: No

Reviewer #3: No

Reviewer #4: No

---

## [Author Response · Author response to Decision Letter 0]

6 Nov 2019

Reviewer #1

Q1: The neonates in this study were placed in a convenient position for observation during data collection, but the fetus will not be so co-operative. Moreover, the yawn of the near term fetus might occasionally be mechanically constrained because of its position. Further, the neonate is yawning in air, while the fetus will be immersed in amniotic fluid. The timings and definitions may need to be adjusted. 

RESPONSE: We thank the reviewer for their appreciation of our work, and we agree with their remarks. However, as we argued for in a newly inserted paragraph in the general discussion (lines 455-478), the mentioned differences between fetuses and newborns should not hinder the applicability of the proposed method to fetuses, especially as yawn is universally recognized as a stereotyped behavioral pattern. Here is the new paragraph addressing the reviewer’s remarks:

“In order to establish this method as a viable option for identifying fetal yawns in clinical and research settings, additional work should be done to test it on fetal behavior. In particular, the suboptimal framing and variable quality that characterize ultrasonographic scans represent a ubiquitous issue in the study of fetal behavior. However, the fact that the proposed method only relies on the timing of mouth openings highlights a key advantage of this approach, that was specifically conceived in order to overcome the issues of identifying a complex facial behavior such as yawn from ultrasonographic scans. Another potential reason for concern regards the different conditions of fetuses compared with preterm neonates; for example, fetuses are immersed in amniotic fluid and the yawn of the near-term fetus might occasionally be mechanically constrained because of its position. However, there is unanimous agreement in the literature on the stability of the yawning behavioral pattern throughout life [12, 17, 21, 22, 23, 24]. Therefore, the advantages of using preterm neonates as a model for training classificatory algorithms far outweigh these potential issues. 

A strength of the current study was the use of the SCPB [25], a rigorous coding system based on Baby FACS [22], to identify yawns. Because Baby FACS coding is based on multiple and redundant cues to each facial action, facial muscle actions can be reliably identified in fetuses [36], preterm and full-term neonates [37], and infants with facial anomalies [38] as well as typically developing infants. And because the basic coding units of Baby FACS are exhaustive and mutually exclusive, any complex facial movement can be precisely and unambiguously identified in terms of combinations and sequences of its constituent facial muscle actions. Therefore, Baby FACS, which has been referred to as the “gold standard” for coding infants’ facial expressions [39], is especially suited for coding yawning, sensory and perceptual responses [37], pain [40], and other fetal, neonatal and infant behaviors that don’t fit simplified templates for a limited set of emotional expressions.”

Q2: The paper would benefit from more data, sufficient to create an independent test set of equal size. The observation that your Model B is the best performing may be a statistical artifact, or a bit of over training, or it may be real. I can't say.

RESPONSE: We fully embrace the reviewer's suggestion and inserted this consideration as a new sentence (lines 441-443, reported below). 

“Further research is needed in order to confirm these results on bigger samples, e.g. regarding the best performing model, as well as to provide additional training for the SVM algorithm.”

Reviewer #2

Q1: In the title it is probably better to replace “preterm model” with “preliminary model” or “initial model”. Unless the adjective “preterm” refers to neonates.

RESPONSE: We thank the reviewer for their appreciation of our work. Preterm indeed refers to neonates: To make it clear we changed the titles as follows: "Identifying fetal yawns based on temporal dynamics of mouth openings: A preterm neonate model using support vector machines (SVMs)".

Q2: How were SVM parameter values determined (C=100, gamma=1, line 316) and what does "type 1 error" and "type 2 error" (lines 313-314) mean?

RESPONSE: SVM parameters were set at the default value for the package adopted. Since results were optimal, and the aim of the Study 2 was to test the feasibility of the proposed SVM-based method, they were reported only for replicability purposes. Type I and II errors mean respectively false positives and negatives. A specification has been introduced (lines 326-327)

Q3: It would be worth describing more precisely how sensitivity and specificity were determined in study 1.

RESPONSE: The methods adopted to calculate sensitivity and specificity were explicitly reported at their first occurrence (lines 237-242, reported below).

“In order to assess the accuracy of the measure adopted by Reissland et al. [10], we used a contingency table to assess its sensitivity (i.e. the true positive rate, calculated as the proportion of yawns that were correctly classified as yawns) and specificity (i.e. the true negative rate, calculated as the proportion of non-yawns that were correctly classified as non-yawns), and compute Cohen’s Kappa.”

Q 4: The abbreviation "AU" is explained later (line 190) than its first use in the text (line 156).

RESPONSE: The "AU" abbreviation was explained at its first occurrence and all abbreviations were checked throughout the manuscript (line 129).

5) What does "t(126)" (lines 346-353) mean?

RESPONSE: The t-statistic is the ratio of the departure of the estimated value of a parameter from its hypothesized value to its standard error. The value in parentheses represents degrees of freedom. 

Q6: English can be slightly improved in some places, e.g.:

Q6.1: “construct validity of the measure” -> “validity of the measure” or “correctness of the measure”

RESPONSE: Construct validity, defined as the degree to which a test measures what it claims, or purports, to be measuring, is one of the key concepts in psychometrics. Because this is technical terminology, we maintained "construct validity of the measure" in the manuscript, after consulting the English mother tongue author.

Q6.2: “risk of sacrificing accuracy for precision” -> “risk of lowering accuracy at the expense of precision” (line 96)

RESPONSE: In our understanding, the reviewer’s proposed sentence does not retain the original meaning, as precision is not affected. In order to make the phrase more immediately understandable to the reader and not redundant, we deleted "for precision" from the original formulation. (line 99)

Q6.3- line 385: “having achieved the highest” -> “having the highest” or “achieving the highest”

RESPONSE: In our understanding, the original wording correctly uses the “perfect participle” tense. “The perfect participle indicates completed action and was therefore maintained in the revised manuscript. (https://www.learnenglish.de/grammar/participleperfect.html) 

Q6.4: line 389: “did not just display a longer” -> “not only showed a longer”

RESPONSE: The suggested version was adopted (line (403)

Q6.5: line 394: “according to one specific iteration” -> “in a given iteration”

RESPONSE: The original wording refers to a specific iteration taken at random, not to any iteration. However, Figure 2 was deleted from the new version of the manuscript because of its limited informative value.

- line 156: “et al” -> “et al.”, “F For” -> “For”, line 383: “011” -> “0.11”

RESPONSE: The suggested revisions were adopted.

Reviewer #3

Q1: The abbreviation usage needs improving. Each abbreviation shall be explained at the first use. The “symbol” t(126) (page 10) needs explanation as well?

RESPONSE: We thank the reviewer for their appreciation of our work. Abbreviations were checked and explained at their first use. The t-statistic is the ratio of the departure of the estimated value of a parameter from its hypothesized value to its standard error. The value in parentheses represents degrees of freedom. We propose to maintain the original, more concise formulation.

Q2: There are some typos, e.g.: “F For” -> “For” (line 156), “011” -> “0.11” (line 383). The text should be checked thoroughly.

RESPONSE: The manuscript was checked for typos and corrected.

Q3: How were SVM parameter values (C and γ) determined?

RESPONSE: SVM parameters were set at the default value for the package adopted. Since results were optimal, and the aim of Study 2 was to test the feasibility of the proposed SVM-based method, they were reported only for replicability purposes. 

Q4: Is the classification performance defined on the testing sets only? If yes, the testing set would contain 5 yawing cases (on average). As the generalization ability estimation is of the crucial importance, is this number enough for making the conclusions about fetal yawn models?

RESPONSE: Classification performance was calculated based on testing sets only, but the hold-out validation included 30 iterations, and for each different training and testing sets were randomly selected. Moreover, both according to previous literature and our results, yawns seem to represent a very specific population of mouth openings, in terms of temporal dynamics. Therefore, we believe that the proposed sample is sufficient to show the feasibility of such method.

Q5: It seems that the figure 2 shows the SVM classification result for Model A (which is not explicitly stated). But what is its purpose? Why plateau duration was fixed at 1.05 s. (half of the mean of plateau duration related to fetal yawns)?

RESPONSE: The purpose of figure 2 was simply to illustrate how SVM models work in establishing hyper-planes for classifying mouth openings. Plateau duration was fixed around the shortest duration coded for yawns, in order to show more informative decision boundaries. The displayed model is Model D, because the adopted R package (e1071) did not allow for graphical representation of interaction-only models. In consideration of these limitations, and of its limited informative content, we propose to delete the figure from the new version of the manuscript.

Q6 As the classification results are so high, did Authors consider to model the fetal yawns by using only two temporal dynamics variables of the highest discrimination ability (like opening and plateau phases – see Figure 1)?

RESPONSE: Other methods were tried out in a preliminary phase. However, the difference between opening and closing (opening/closing asymmetry) also had high discrimination ability, and is directly related to the description, ubiquitous in the pertinent literature, of yawning as characterized by a long opening and a short closing phase.

Reviewer #4

Q1: Introduction. Regarding the difference among reported yawns frequencies, which is the role of circadian changes in yawns activity of the fetus. Does yawns activity change over time? I would suggest discussing this point.

RESPONSE: We thank the reviewer for their appreciation of our work and the valuable comments and suggestions. We inserted a new paragraph in the introduction (lines 54-59) in order to briefly discuss the potential effects of circadian rhythms and other variables (see below).

“The sharp differences between these results might be partially explained by different factors, including fetal circadian rhythms and pathological conditions [2, 3]. However, all studies included healthy fetuses and most had US scans performed during the afternoon [11, 12, 13, 14, 15, 18, 19]. Therefore, the inconsistencies shown in Table 1 suggest that the measures used in these studies lacked adequate reliability or validity.

Q2: The Authors did not mention the sample size calculation for their study. I would suggest reporting how the Authors chosen the number of included neonates.

RESPONSE: Because yawn is universally known as stereotyped behavioral pattern (see Table 1), we concluded that even a small sample (130 mouth openings) would be sufficient for our purposes, i.e. to assess the validity of the method proposed by Reissland and colleagues (in Study 1) and to test the feasibility of more sophisticated SVM-based methods for identify yawns (in Study 2). The results of Study 2 confirm this assumption; models highlighted good performances in distinguishing yawns and non-yawns. It is noteworthy that preterm neonates, despite representing a good model for fetal behavior, represent a very difficult population to sample and study. Because, in our experience, sample size calculation is generally not reported in studies adopting SVM or other kinds of classification algorithms, and classification performance is generally intended as the criterion for determining the required sample size for training (see https://www.ncbi.nlm.nih.gov/pmc/articles/PMC3307431/), we propose not to introduce this specification.

Q3: The authors have not adequately highlighted the strengths and limitations of their study. I suggest better specifying these points. The Authors use the Baby FACS coders as the gold standard for the evaluation of yawns, is it actually the gold standard? – The newborns yawns in air and fetus yawns in water, do Authors expect some changes?

RESPONSE: We fully embrace the reviewer's suggestions and inserted the pertinent considerations as a new paragraph in the discussion (lines 455-478, reported below).

“In order to establish this method as a viable option for identifying fetal yawns in clinical and research settings, additional work should be done to test it on fetal behavior. In particular, the suboptimal framing and variable quality that characterize ultrasonographic scans represent a ubiquitous issue in the study of fetal behavior. However, the fact that the proposed method only relies on the timing of mouth openings highlights a key advantage of this approach, that was specifically conceived in order to overcome the issues of identifying a complex facial behavior such as yawn from ultrasonographic scans. Another potential reason for concern regards the different conditions of fetuses compared with preterm neonates; for example, fetuses are immersed in amniotic fluid and the yawn of the near-term fetus might occasionally be mechanically constrained because of its position. However, there is unanimous agreement in the literature on the stability of the yawning behavioral pattern throughout life [12, 17, 21, 22, 23, 24]. Therefore, the advantages of using preterm neonates as a model for training classificatory algorithms far outweigh these potential issues. 

A strength of the current study was the use of the SCPB [25], a rigorous coding system based on Baby FACS [22], to identify yawns. Because Baby FACS coding is based on multiple and redundant cues to each facial action, facial muscle actions can be reliably identified in fetuses [36], preterm and full-term neonates [37], and infants with facial anomalies [38] as well as typically developing infants. And because the basic coding units of Baby FACS are exhaustive and mutually exclusive, any complex facial movement can be precisely and unambiguously identified in terms of combinations and sequences of its constituent facial muscle actions. Therefore, Baby FACS, which has been referred to as the “gold standard” for coding infants’ facial expressions [39], is especially suited for coding yawning, sensory and perceptual responses [37], pain [40], and other fetal, neonatal and infant behaviors that don’t fit simplified templates for a limited set of emotional expressions.”

Q4: As appropriately stated by the authors, fetal yawning has been associated with proper neurological development. In this regard, the possibility to detect these fetal movements may play a pivotal importance as additional tool to assess fetal wellbeing in utero. I would suggest stressing the possible underlying causes of fetal neurological impairment, such as for example placental diseases (refer to: PMID: 28282763; PMID: 28466013; PMID: 28243732).

RESPONSE: We thank the reviewer for this important remark. References to this aspect were inserted at the beginning of the discussion and at the end of the discussion (respectively lines 414-417 and 493-496, reported below).

“The ability to identify yawns in fetuses and to distinguish yawns from non-yawn mouth openings is of interest because accurate detection of this widely observed behavior can have potential clinical importance for identifying early signs of neurodevelopmental abnormalities and other conditions such as placental diseases [35] in fetuses, very early preterm infants, and other populations at risk.”

“In conclusion, the development of a reliable and valid method for identifying yawning can provide a potentially valuable tool for studying perinatal behavior and for assessing fetal and preterm infant wellbeing. Future research using machine-learning systems could contribute to the development of sensitive measures for early diagnosis of neurological impairments and other disorders.”

---

## [Decision Letter · Decision Letter 1]

2 Dec 2019

PONE-D-19-18762R1

Identifying fetal yawns based on temporal dynamics of mouth openings: A preterm neonate model using support vector machines (SVMs)

PLOS ONE

Dear Prof. Dondi,

Thank you for submitting your manuscript to PLOS ONE. After careful consideration, we feel that it has merit but does not fully meet PLOS ONE’s publication criteria as it currently stands. Therefore, we invite you to submit a revised version of the manuscript that addresses the points raised during the review process.

The Reviewers appreciated the manuscript improvements. Only one point was questioned by Reviewer 3. I would suggest addressing this point or introducing a comment in the methods explaining the chosen method instead of the recommended as an answer to the Reviewer. Further details are provided in the Reviewer's recommendations. 

We would appreciate receiving your revised manuscript by Jan 16 2020 11:59PM. To enhance the reproducibility of your results, we recommend that if applicable you deposit your laboratory protocols in protocols.io, where a protocol can be assigned its own identifier (DOI) such that it can be cited independently in the future. For instructions see: http://journals.plos.org/plosone/s/submission-guidelines#loc-laboratory-protocols

We look forward to receiving your revised manuscript. 

Kind regards,

Simone Garzon

Academic Editor

PLOS ONE

Reviewers' comments:

Reviewer's Responses to Questions

**Comments to the Author**

1. If the authors have adequately addressed your comments raised in a previous round of review and you feel that this manuscript is now acceptable for publication, you may indicate that here to bypass the “Comments to the Author” section, enter your conflict of interest statement in the “Confidential to Editor” section, and submit your "Accept" recommendation.

Reviewer #2: All comments have been addressed

Reviewer #3: All comments have been addressed

Reviewer #4: All comments have been addressed

2. Is the manuscript technically sound, and do the data support the conclusions?

Reviewer #2: (No Response)

Reviewer #3: Yes

Reviewer #4: Yes

3. Has the statistical analysis been performed appropriately and rigorously? 

Reviewer #2: (No Response)

Reviewer #3: No

Reviewer #4: Yes

4. Have the authors made all data underlying the findings in their manuscript fully available?

Reviewer #2: (No Response)

Reviewer #3: Yes

Reviewer #4: Yes

5. Is the manuscript presented in an intelligible fashion and written in standard English?

Reviewer #2: (No Response)

Reviewer #3: Yes

Reviewer #4: Yes

6. Review Comments to the Author

Reviewer #2: (No Response)

Reviewer #3: R3, Q3: I cannot agree that default values of SVM parameters ensure optimal classification results. In my opinion, for each of the considered models it is necessary to experimentally find C and γ values ensuring the best model performance. For example the first 10 iterations of the hold-out validation can be used to find the parameters (from a reasonable range of C and γ) providing highest values of performance indices. Since it may be difficult to analyze several performance indicators simultaneously, F-Score (or similar) can be employed. However, even in this case the claim of the C and γ optimality is questionable.

Reviewer #4: Authors have performed the required changes, improving significantly the quality of the article.

I have no further concerns.

7. PLOS authors have the option to publish the peer review history of their article (what does this mean?). If published, this will include your full peer review and any attached files.

Reviewer #2: No

Reviewer #3: No

Reviewer #4: No

---

## [Author Response · Author response to Decision Letter 1]

4 Dec 2019

Reviewer #3: R3, Q3: I cannot agree that default values of SVM parameters ensure optimal classification results. In my opinion, for each of the considered models it is necessary to experimentally find C and γ values ensuring the best model performance. For example the

first 10 iterations of the hold-out validation can be used to find the parameters (from a reasonable range of C and γ) providing highest values of performance indices. Since it may be difficult to analyze several performance indicators simultaneously, F-Score (or similar) can be employed. However, even in this case the claim of the C and γ optimality is questionable.

RESPONSE: We really thank the reviewer for insisting on this important point. As suggested, we adopted grid search to optimize C and γ values (see lines 329-332, see below).

“Grid search based on 10-fold cross-validation error was employed to optimize the parameters C and gamma using the function “tune.svm”, within the interval [10-3, 103]. Model A was optimized with C = 100 and gamma = 0.1 while Model B was optimized with C = 10 and gamma = 1.”

Moreover, because once we optimized these parameters within the interval [10-3, 103] the full model (originally model E, now model A) highlighted the best performance (as is generally expected), we simplified the SVM analysis by deleting three models and retaining only the full model (Model A) and Model B (including the three-ways interaction of total duration, plateau duration and opening/closing asymmetry). The rationale for maintaining Model B is that it shows that the interaction of these three variables is responsible for a significant part of the ability to discriminate between yawns and non-yawn mouth openings. On the other hand, because the full model is now the best performing one and differences between models are smaller once C and gamma are optimized, we think that including all five original models would only contribute to make results less clear. Methods, results and discussion of Study 2 were modified accordingly.

---

## [Decision Letter · Decision Letter 2]

10 Dec 2019

Identifying fetal yawns based on temporal dynamics of mouth openings: A preterm neonate model using support vector machines (SVMs)

PONE-D-19-18762R2

Dear Dr. Dondi,

We are pleased to inform you that your manuscript has been judged scientifically suitable for publication and will be formally accepted for publication once it complies with all outstanding technical requirements.

With kind regards,

Simone Garzon

Academic Editor

PLOS ONE

Additional Editor Comments (optional):

Reviewers' comments:

Reviewer's Responses to Questions

**Comments to the Author**

1. If the authors have adequately addressed your comments raised in a previous round of review and you feel that this manuscript is now acceptable for publication, you may indicate that here to bypass the “Comments to the Author” section, enter your conflict of interest statement in the “Confidential to Editor” section, and submit your "Accept" recommendation.

Reviewer #3: All comments have been addressed

2. Is the manuscript technically sound, and do the data support the conclusions?

Reviewer #3: Yes

3. Has the statistical analysis been performed appropriately and rigorously? 

Reviewer #3: Yes

4. Have the authors made all data underlying the findings in their manuscript fully available?

Reviewer #3: Yes

5. Is the manuscript presented in an intelligible fashion and written in standard English?

Reviewer #3: Yes

6. Review Comments to the Author

Reviewer #3: (No Response)

7. PLOS authors have the option to publish the peer review history of their article (what does this mean?). If published, this will include your full peer review and any attached files.

Reviewer #3: No

---

## [Editor Report · Acceptance letter]

11 Dec 2019

PONE-D-19-18762R2 

Identifying fetal yawns based on temporal dynamics of mouth openings: A preterm neonate model using support vector machines (SVMs) 

Dear Dr. Dondi:

I am pleased to inform you that your manuscript has been deemed suitable for publication in PLOS ONE. Congratulations! Your manuscript is now with our production department. 

With kind regards,

on behalf of

Dr. Simone Garzon 

Academic Editor

PLOS ONE